# Tumor-Infiltrating Lymphocytes (TILs) in Early Breast Cancer Patients: High CD3^+^, CD8^+^, and Immunoscore Are Associated with a Pathological Complete Response

**DOI:** 10.3390/cancers14102525

**Published:** 2022-05-20

**Authors:** Bernardo Leon Rapoport, Simon Nayler, Bernhard Mlecnik, Teresa Smit, Liezl Heyman, Isabelle Bouquet, Marine Martel, Jérôme Galon, Carol-Ann Benn, Ronald Anderson

**Affiliations:** 1Department of Immunology, Faculty of Health Sciences, University of Pretoria, Pretoria 0001, South Africa; liezlheyman@gmail.com (L.H.); ronald.anderson@up.ac.za (R.A.); 2The Medical Oncology Centre of Rosebank, Johannesburg 2193, South Africa; data@rapoport.co.za; 3Breast Care Centre, Milpark Hospital, Johannesburg 2193, South Africa; simonn@histologic.co.za (S.N.); drbenncarol@gmail.com (C.-A.B.); 4Gritzman and Thatcher Inc. Laboratories, Johannesburg 2193, South Africa; 5Donald Gordon Medical Centre, University of the Witwatersrand, Johannesburg 2193, South Africa; 6Inovarion, 75005 Paris, France; bernhard.mlecnik@inovarion.com; 7Veracyte, Luminy Biotech Entreprises, 13009 Marseille, France; Isabelle.Boquet@haliodx.com (I.B.); marine.martel@haliodx.com (M.M.); jerome.galon@crc.jussieu.fr (J.G.); 8Laboratory of Integrative Cancer Immunology, INSERM, 75005 Paris, France; 9Equipe Labellisée Ligue Contre le Cancer, 75005 Paris, France; 10Centre de Recherche des Cordeliers, Sorbonne Université, 75005 Paris, France; 11Helen Joseph Hospital Breast Centre, Johannesburg 2193, South Africa

**Keywords:** CD3^+^ lymphocytes, CD8^+^ lymphocytes, early breast cancer, HER2-positive breast cancer, Ki-67, luminal breast cancer, TILs

## Abstract

**Simple Summary:**

In 2021, the World Health Organization announced that breast cancer had overtaken lung cancer to become the most common cancer globally, accounting for 12% of all new cancer cases, with younger women resident in low-income countries having the lowest 5-year survival rates. The main aim of the current study was to evaluate the prognostic utility of an innovative, objective, computer-assisted, digital imaging procedure known as the Immunoscore for clinical research (ISCR) as a strategy to reveal the efficiency of the anti-tumor cellular immune landscape of the tumor microenvironment (TME) in biopsies taken from women diagnosed with early breast cancer prior to administration of neoadjuvant chemotherapy followed by surgical resection. Our results demonstrated the ability of the ISCR to enumerate tumor-infiltrating lymphocytes in the TME and, in particular, to illustrate the spatial arrangement of these cells, which, importantly, correlated with clinical outcome, measured as the pathological complete response.

**Abstract:**

Background: Tumor-infiltrating lymphocytes are associated with a better prognosis in early triple-negative breast cancer (TNBC). These cells can be enumerated in situ by the “Immunoscore Clinical Research” (ISCR). The original Immunoscore^®^ is a prognostic tool that categorizes the densities of CD3^+^ and CD8^+^ cells in both the invasive margin (IM) and center of the tumor (CT) in localized colon cancer, yielding a five-tiered classification (0–4). We evaluated the prognostic potential of ISCR and pathological complete response (pCR) following neoadjuvant chemotherapy (NACT). Methods: The cohort included 53 TNBC, 32 luminal BC, and 18 HER2-positive BC patients undergoing NACT. Pre-treatment tumor biopsies were immune-stained for CD3^+^ and CD8^+^ T-cell markers. Quantitative analysis of these cells in different tumor locations was performed using computer-assisted image analysis. Results: The pCR rate was 44%. Univariate analysis showed that primary tumor size, estrogen-receptor negative, progesterone-receptor negative, luminal vs. HER2-positive vs. TNBC, high Ki-67, high densities (cells/mm^2^) of CD3 CT, CD8^+^ CT, CD3^+^ IM, and CD8^+^ IM cells were associated with a high pCR. ISCR was associated with pCR following NACT. A multivariate model consisting of ISCR and the significant variables from the univariate analysis showed a significant trend for ISCR; however, the low sample size did not provide enough power for the model to be included in this study. Conclusions: These results revealed a significant prognostic role for the spatial distributions of the CD3^+^, and CD8^+^ lymphocytes, as well as the ISCR in relation to pCR following NACT.

## 1. Introduction

Cancer cells interact with and modulate the immune system, leading to an imbalance between tumor growth and host surveillance. Increasing literature has shown that the tumor microenvironment (TME) plays an essential role in disease progression, with the presence of tumor-infiltrating-lymphocytes (TILs) being associated with a favorable prognosis in various types of cancers [1,2]. The quantification of TILs in formalin-fixed tumor samples ranges from hematoxylin/eosin (H&E) staining-based visual scoring of lymphocytic infiltrates to multiplex immunohistochemistry and immunofluorescence. In the case of breast cancer (BC), it has been recognized for some time that the presence of TILs in the TME is associated with a better outcome, particularly in those patients with triple-negative breast cancer (TNBC). This clearly underscores the benefit of immune checkpoint blockade treatments via monoclonal antibody (Mab)-based targeting of PD-1/PD-L1 with agents such as atezolizumab and pembrolizumab [3,4,5,6].

The current staging system is based on the TNM stage (tumor (T), node (N), and metastasis (M)) that has been used for many years [7]. However, patients with the same TNM stage can have different clinical outcomes following neoadjuvant chemotherapy. In this context, a significant shortcoming of the TNM classification is that it does not consider the patient anti-tumor immune response. This limitation may be overcome by combining clinical cancer staging procedures with an objective, automated, prognostic immune scoring system. The Immunoscore^®^ (IS^®^) represents a classification system that incorporates the number, type, and distribution of immune cells [8,9,10] using the CD3^+^ and CD8^+^ cell densities in the center of the tumor and the invasive margin of localized colon cancer patients, with higher scores indicative of infiltration by more significant numbers of total T cells (CD3^+^) and cytotoxic T cells (CD8^+^) in both the tumor core and margins. International validation of the prognostic potential of the consensus IS^®^ has been convincingly demonstrated in early colon cancer [10,11], and this digital pathology diagnostic assay has recently been incorporated into the localized colon cancer ESMO Guidelines [12,13]. Based on the colon cancer data, the current study was undertaken to evaluate the IS^®^ for clinical research (ISCR) in patients with early BC and the association with a pathological complete response (pCR) following neoadjuvant chemotherapy (NACT). The ISCR was calculated according to the densities of the CD3^+^ and CD8^+^ cells in the center of the tumor (CT) and invasive margin (IM) regions. The receiver operating characteristic (ROC) curve method determined a cut-off point for CD3^+^ IM, CD3^+^ CT, CD8^+^ IM, and CD8^+^ CT and was classified as high or low for each variable. If both markers were elevated in the CT and IM regions, the highest ISCR score of 4 was given. If one marker was high in the CT, but low in the IM, while the other was high in both regions, an ISCR of 3 was assigned. A similar method was applied to scores of 2 and 1. Lastly, the lowest possible score (0) was assigned if both markers were low in both regions (refer to Materials and Methods for details).

## 2. Materials and Methods

### 2.1. Patient Population

Pre-treatment tumor blocks were retrieved from 103 consecutive breast cancer patients who underwent neoadjuvant chemotherapy (NACT) between December 2004 and May 2019. All patients who completed NACT and had surgery were eligible for the study. Patients with non-metastatic BC undergoing NACT were included, and their characteristics are described in Table 1.

Clinical data included tumor stage, NACT, surgery, date of recurrence, and death date, which were collected retrospectively. All patients were treated with a standard anthracycline regimen and/or taxane-based NACT. HER2-positive patients also received trastuzumab. It is an institutional policy to use NACT for most patients with HER2-positive and TNBC.

### 2.2. Immunohistochemistry (IHC) and Digital Image Analysis

For each patient, a pre-treatment paraffin-embedded tumor block from the Center of the Tumor (CT) and the Invasive Margin (IM) was selected. Two tissue paraffin sections of 4 microns (µm) were processed for IHC staining for CD3 and CD8 antibodies according to a protocol optimized by HalioDX (a Veracyte Company) and previously described by [10]. Digital slides were obtained with a 20 × magnification and a resolution at 0.45 µm/pixel. Image analysis was performed on the scanned digital slides. The mean CD3 and CD8 cell densities were determined in the CT and IM regions using a specially developed Immunoscore^®^ module (INSERM, Paris, France) integrated into the image analysis system Developer XD (Definiens, Munich, Germany).

### 2.3. Pathological Assessment

Immunohistochemical staining was performed for estrogen receptors (ER), progesterone receptors (PR), and HER2 positivity, as well as the Ki-67 proliferative index. Fluorescence in situ hybridization (FISH) was used to confirm HER2 positivity. Patients who completed NACT were eligible (*n* = 103) for the ISCR analysis. Clinical data were collected from medical records. Pathological complete response (pCR) was defined as the complete disappearance of invasive cancer in the breast and the absence of tumor in the axillary lymph nodes.

### 2.4. Immunoscore^®^ for Clinical Research

The Immunoscore^®^ assay is the first standardized and commercialized immune-based assay for prognosis in early-stage colon cancer [14]. Immunoscore^®^ is a registered trademark of Inserm. It assesses the host immune response by measuring intra- and peri-tumoral T-cell infiltration in formalin-fixed paraffin-embedded (FFPE) tissue sections. Initially developed for the prognosis of colon cancer, it is now widely used in clinical research for prognostic and predictive evaluation in many types of solid tumors within drug development programs (ISCR).

As a first clinical validation of the prognostic potential of the ISCR in patients with BC, we assessed this test in BC in the neoadjuvant setting.

The ISCR was determined based on the CD3 and CD8 cell densities from the paraffin-embedded tumor block. If both markers were elevated in the CT regions, a score of 2 (High) was given. If only one marker was high in the CT and the other one low, a score of 1 (Intermediate) was assigned, while if both markers were low in the CT, then a score of 0 (Low) was allocated. Because it was previously shown that biopsy simulations could accurately estimate intra-metastatic immune infiltrates [15], we are confident that the estimated density closely reflects the reality of the infiltrate.

### 2.5. Statistical Methods

Descriptive statistics were used to tabulate patient characteristics. Associations of clinical and pathological characteristics, including Ki-67 and CD3^+^ and CD8^+^ cytotoxic T cells with pCR, were determined. As mentioned above, receiver-operating characteristic (ROC) curves with analysis of areas under the curves (AUC) were used to determine the optimal cut-off points for Ki-67, CD3^+^, CD8^+^ cytotoxic T cells, and the ISCR. The Youden index, a summary measure of the ROC curve, was used as an agnostic method for choosing an optimal cut-off value. Chi-square and Fisher exact tests were used to compare categorical data. The Mann–Whitney U test was used for continuous variables. A univariate logistic regression model was used to compare categorical variables. Multivariate analysis using a logistic regression multivariate model was used, which included only variables that exhibited a univariate association with the dependent variable, pCR (*p* < 0.1). A *p*-value of <0.05 was considered statistically significant. NCSS 2021 software for Windows (USA) was used for statistical analyses.

### 2.6. Ethics Approval

This study investigated the role of ISCR and pCR rate in103 patients with early breast cancer undergoing NACT. Institutional ethics approval was obtained from the Human Sciences Research Council (HSRC) of South Africa (Protocol No: 17051653/Breast 001).

## 3. Results

We performed the ISCR in 103 BC patients who received neoadjuvant anthracycline- and/or taxane-based (with/without trastuzumab) chemotherapy (Table 1). Patients had a median age of 52 years (ranging from 26–84 years). The biological sub-types of BC included 53 TNBC patients (52%), 32 luminal patients (31%), and 18 HER2-positive patients (17%). Pre-menopausal and post-menopausal patients numbered 41 (40%) and 62 (60%), respectively. The tumor size was categorized as follows: T1 patients–23 (22%), T2 patients 65 (63%), and T3 & T4 combined 15 (15%) patients. Of these patients, 54 (52%) had nodal involvement. The pCR rate for the entire cohort was 44% (luminal 9%, TNBC 62%, and HER2-positive 50%). 

As shown in Table 2, the median cell density counts for the centers of the tumors were CD3^+^ 884 cells/mm^2^ (25–5771 cells/mm^2^) and CD8^+^ 358 cells/mm^2^ (10–3448 cells/mm^2^). The median invasive margin cell densities for CD3^+^ were 1409 cells/mm^2^ (53–6197 cells/mm^2^) and CD8^+^ 535 cells/mm^2^ (38–311 cells/mm^2^).

### 3.1. Receiver Operating Characteristic (ROC) Curve for CD3^+^ and CD8^+^ Cells in the Center of the Tumor and the Invasive Margin

The discriminatory potential of the prognostic biomarkers was investigated using a binary classification model for attaining a pCR. An ROC curve for CD3^+^ and CD8^+^ cells in the center of the tumor and the invasive margin is shown in Figure 1. As shown in Figure 1, the sensitivity and 1-specificity values from each cut-off were plotted as ROC curves.

The following cut-off levels were obtained:CD3^+^ T-cell densities in the center of the tumor were 1186.80 cells/mm^2^ (sensitivity 79.3% and specificity 64.4%)CD8^+^ T-cell densities in the center of the tumor were 324,50 cells/mm^2^ (sensitivity 77.8%% and specificity 67.2%)CD3^+^ T-cell densities in the invasive margin were 1093.70 cells/mm^2^ (sensitivity 93.5% % and specificity 63.9%)CD8^+^ T-cell densities in the invasive margin were 431.5 cells/mm^2^ (sensitivity 87.1% and specificity 65.9%).

### 3.2. Categorization of Patients According to Numbers of CD3^+^ and CD8^+^ Cells in the Center of the Tumor and Invasive Margin

The CD3^+^ IM and CD8^+^ IM were not detected in 28 patients. The full set of CD3^+^ CT, CD3^+^ IM, CD8^+^ CT and CD8^+^ IM was detected in 75 patients. There were 40 (39%) and 63 (61%) patients with CD3^+^ cell counts in the center of the tumor of ≥1186.81 cells/mm^2^ and <1186.80 cells/mm^2^, respectively. Additionally, 46 (61%) and 29 (39%) patients had CD3^+^ cell counts in the invasive margin of ≥1093.70 cells/mm^2^ and <1093.71 cells/mm^2^, respectively. Fifty-five (53%) and 48 patients (47%) had CD8^+^ cell counts in the center of the tumor of ≥324.51 cells/mm^2^ and <324.50 cell/mm^2^, respectively. Forty-one (55%) and 34 (45%) of patients had CD8^+^ cell counts in the invasive margin of ≥431.51 cells/mm^2^ and <431.50 cells/mm^2^, respectively (Table 3). Of the patient population, 36% and 31% had ISCR values of 0 and 4, respectively. See Table 3.

An analysis of the characteristics of the 75 patients with a documented IM score relative to the 28 patients without a documented IM score was undertaken to detect potential differences between the two groups. Among the 75 patients with known IM scores, there were 40 (53%) TNBC patients, 23 (31%) luminal patients, and 12 (16%) HER2-positive patients. When compared to the distribution of the molecular subtypes of the entire cohort of 103 patients with the 75 patients with known IM ISCR, we did not detect any significant difference between the two groups (TNBC 51% vs. 53%, luminal subtype 31% vs. 31% and HER2-positive 18% vs. 16%; χ^2^ = 0.547, DF = 2, *p* < 0.7607).

No significant difference in terms of menopausal status (pre-menopausal vs. post-menopausal; χ^2^ = 0.004, *p* < 0.9475), nodal status (negative vs. positive; χ^2^ = 1.001, *p* < 0.3170), stage (stage I vs. IIA vs. IIB vs. III; χ^2^ = 2.265, *p* < 0.5192) or molecular sub-type (TNBC vs. luminal vs. HER2 status; χ^2^ = 0.547, *p* < 0.7607) was found between the two cohorts, with the exception of tumor size (T1 vs. T2 vs. T3 combined with T4; χ^2^ = 9.358, *p* < 0.0093).

### 3.3. Comparison of T-Cell Density between TNBC vs. Non-TNBC Patients

As shown in Figure 2, T-cell subset densities (CD3^+^ and CD8^+^ in the center of the tumor and invasive margin) were significantly higher in TNBC vs. non-TNBC patients.

### 3.4. Pathological Complete Response According to CD3^+^ and CD8^+^ Cell Counts in the Center of the Tumor and the Invasive Margin

The median CD3^+^ and CD8^+^ cells/mm^2^ in the center of the tumor and the invasive margin were significantly higher for patients who attained a pCR than for patients who did not. The median CD3^+^ cell count in the center of the tumor for patients who attained a pCR was 1432 cells/mm^2^ compared to those patients who did not achieve a pCR, with a median cell count of 567 cells/mm^2^ (*p* < 0.0033). The median CD3^+^ density in the invasive margin was 1877 cells/mm^2^ for patients who attained a pCR, compared to a median of 540 cells/mm^2^ for those patients who did not attain a pCR (*p* < 0.0004). Additionally, the median CD8^+^ cell count in the center of the tumor for patients who attained a pCR was 614 cells/mm^2^ compared to those patients who did not achieve a pCR with a median cell count of 246 cells/mm^2^ (*p* < 0.0199). The median CD8^+^ cell count in the invasive margin was 827 cells/mm^2^ for patients attaining a pCR compared to a median of 255 cells/mm^2^ for those patients without a pCR (*p* < 0.0012). The proportion of patients attaining a pCR with a high CD3 CT was 71%, compared to 29% pCR for those with a low CD3 CT (χ^2^ = 12.7625, *p* < 0.0004). Additionally, the pCR rate of patients with a high CD8 CT was 68% compared to 32% with a low CD8 CT (χ^2^ = 14.2535, *p* < 0.0002). For patients with a high CD3 IM, the pCR was 64% compared to 35% for those with a low CD3 IM (χ^2^ = 15.1301, *p* < 0.0001). For those patients with a high CD8 IM, the pCR rate was 64%, in comparison with 36% for those with a low CD8 IM (χ^2^ = 17.4957, *p* < 0.0003). See Figure 3 and Figure 4. 

### 3.5. Pathological Complete Response by Immunoscore^®^ Clinical Research

As shown in Table 4, CD3^+^ and CD8^+^ IM values were not detected in 28 patients. The full set of CD3^+^ CT, CD3^+^ IM, CD8^+^ CT and CD8^+^ IM was detected in 75 patients. The pCR rate was significantly different in those patients with a high ISCR (3 and 4 = 69%) relative to those with intermediate (2 = 50%) and low (0 and 1 = 6%) ISCR values (*p* = 0.006).

### 3.6. Logistic Regression Analysis for Patient Characteristics Associated with a Pathological Complete Response

As shown in Table 5, the pCR rate of the entire cohort was 44%. On univariate logistic regression analysis, factors associated with a higher pCR included primary tumor size ≤ 2 cm vs. tumor size 2 cm to 5 cm (*p* < 0.0202) and primary tumor size ≤ 2 cm vs. tumor size > 5 cm (*p* < 0.0102), nodal status positive vs. negative (*p* < 0.0246). Additionally, other significant variables included ER status positive vs. negative (*p* < 0.0001), PR status positive vs. negative (*p* < 0.0001), molecular subtype, luminal vs. HER2-positive (*p* < 0.0001) and TNBC vs. HER2-positive (*p* < 0.0052), Ki-67 >40% vs. 15% to 39% (*p* < 0.0001), and Ki-67 <15% vs. >40% (*p* < 0.0052) and stage IIB and stage III vs. stage I and stage IIA (*p* < 0.0001) (see Table 5). An exploratory analysis was done to detect potential differences in the ISCR in relation to the treatment administered (anthracycline alone = 2 vs. taxane alone = 13 vs. combination = 88).

Additionally, 18 HER2-positive patients received trastuzumab. Due to the small number of patients (who were treated with anthracycline alone or taxane alone), no meaningful significance could be detected. The ISCR had a significant trend in a multivariate model, which included significant variables from univariate analysis, but the low sample size in the current study did not provide enough power for the model to be applied.

### 3.7. Positive Predictive Values and Negative Predictive Values for CD3^+^ IM, CD8^+^ IM, CD3^+^ CT, CD8^+^ CT, and ISCR for pCR

As shown in Table 6, the positive predictive value (PPV) for ISCR (ISCR 0-1 vs. ISCR 2-3-4) was 60.4% (95% CI = 45.3–74.2), compared to 63.2% (95% CI = 46.0–78.2) for CD3^+^ IM, 63.4% (95% CI = 46.9–77.9) for CD8^+^ IM, 62.2% (95% CI = 44.8–77.5) for CD3^+^ CT, and 66.7% (95% CI = 49.0–81.4) for CD8^+^ CT. 

The negative predictive value (NPV) for ISCR (ISCR 0-1 vs. ISCR 2-3-4) was 92.6% (95% CI = 75.7–99.1), compared to 81.1% (95% CI = 64.8–92.0) for CD3^+^ IM, 85.3% (95% CI = 68.9–95.1) for CD8^+^ IM, 79.0% (95% CI = 62.7–90.4) for CD3^+^ CT, and 79.0% (95% CI = 62.7–90.4) for CD8^+^ CT.

## 4. Discussion

Since the beginning of the twentieth century, the presence of immune cell infiltration in various types of cancer was believed to be a favorable prognostic factor associated with a better outcome [1,2]. Although Paul Ehrlich initially proposed the concept of immune surveillance in 1909 [1], the presence of immune cells was not used in cancer staging or understanding clinical outcomes, primarily due to the limited knowledge of the function and composition of the immune system [2]. A century later, experimental animal models established that the immune system was an effective suppressor of cancer development and progression. These experiments demonstrated that immunodeficient mice developed tumors earlier and more frequently than wild-type mice in similar conditions [16]. Subsequently, Pagès et al. examined the role of tumor-infiltrating immune cells in patients with an early metastatic invasion of colorectal cancer; these investigations revealed that the presence of a high-density of infiltrating memory and effector memory T cells (CD45RO+) was associated with a lower incidence of lymphovascular and perineural structures, as well as dissemination of regional lymph nodes [17].

The Immunoscore^®^ was developed to evaluate the presence of immune infiltrates in the TME of colon cancer patients and to enable the assessment of the density, location, and types of different immune infiltrate cells such as total T cells (CD3^+^), cytotoxic T cells (CD8^+^) in the CT, and the IM. The scoring system ranged from Immunoscore^®^ = 0, with low densities of both cell types in the CT and the IM, to Immunoscore^®^ = 4, having high densities of both cell populations in both locations [18].

The current study explored the potential prognostic role of the ISCR in 103 patients with early BC undergoing NACT. We found that a high ISCR was associated with a pCR rate following NACT. Although a considerable amount of earlier work was focused on the enumeration of TILs in breast cancer, to our knowledge, our study is the first to differentiate CT and IM when analyzing the influence of TILs on prognosis after NACT in early BC. In this context, we demonstrated that the numbers of CD3^+^ and CD8^+^ cells in the IM and the CT are robust, independent predictors of pCR in patients with early BC undergoing NACT. In addition, our data showed that the density of cytotoxic lymphocytes at the IM seems to have a similar predictive value to the numbers of those in the CT. This illustrates that lymphocyte distribution within the tumor mass might be more important than just the presence of TILs.

Moreover, our data show a linear relationship between the ISCR and the pCR following NACT in these patients. It is important that, although the study shows a statistical significance for ISCR in predicting pCR post NACT, there are some limitations, including sample size, statistical power, patient selection, and the retrospective nature of this data set. This hypothesis should be confirmed with adequately designed prospective studies, primarily including HER2-positive and TNBC patients, as well as investigating the differences between lymphocyte subsets and TILs. In addition, the utility of the ISCR in predicting responses to immunotherapy (such as PD-L1 inhibitors) in BC in the neoadjuvant setting remains unknown. Furthermore, the positive predictive values and negative predictive values of ISCR, CD3, and CD8 cell densities should also be evaluated in the future, in adequately sized prospective studies.

Several studies have been conducted looking at the prognostic and predictive value of the CD3^+^ and/or CD8^+^ cells with or without immunosuppressive cells in the TME in patients with BC; however, none of these studies investigated the ISCR and its clinical utility [19,20,21,22,23,24,25]. Importantly, ISCR quantification is a reproducible and objective method, whereas the density of TILs is highly subjective and less reproducible [8]. Those differences also reflect the fact that H&E staining of TILs gives a crude and subjective semi-quantitative evaluation of undefined cell populations with divergent functions (e.g., CD4^+^ T cells with Th1 orientation vs. Th2 orientation vs. immune cells with regulatory functions (Treg cells), natural killer (NK) cells, NK T cells, B-cell subsets, innate lymphoid cells, or cytotoxic CD8^+^ T cells) [26,27]. Because of their stability and staining quality, CD3^+^ and CD8^+^ cells were also selected as biomarkers [28]. Our data showed that the ISCR separated patients into subcategories (high vs. low, or high vs. intermediate vs. low) with significant differences with respect to pCR post-NACT.

TILs in the tumor and the adjacent microenvironment indicate a host anti-tumor immune response in patients with BC, which is significantly influenced by the molecular subtype of the tumor. TNBC and HER2-positive BC are more frequently infiltrated by higher levels of TILs, compared to hormone receptor-positive tumors [29,30]. In this context, the study by Denkert et al. conducted by The German Breast Cancer Group is particularly noteworthy. It is the largest study encompassing TILs in various subsets of breast patients (*n* = 3771) in the neoadjuvant setting. Although impressive, despite the large number of patients investigated, these authors reported only on TILs and not on the lymphocyte subsets or spatial distributions [29].

The German Breast Cancer Group evaluated the number of stromal TILs assessed by standardized methods according to the guidelines of the International TIL Working Group. The study included 3771 patients from six prior randomized clinical trials conducted by the same group. TILs were evaluated as continuous variables, as well as three pre-specified groups. These groups consisted of low (0–10% TILs in stromal tissue within the tumor), intermediate (11–59%), and high TILs (≥60%). The investigators performed univariate and multivariate statistical models to evaluate the association between TILs and pCR in these patients.

In patients with hormone receptor-positive, HER2-negative BC subtype, a pCR was attained in 6% of patients with low TILs, 11% with intermediate TILs, and 28% of patients with high TILs levels. In the HER2-positive disease, a pCR was seen in 32% of patients with low TILs, 39% with intermediate TILs, and 48% with high TILs. Lastly, in the TNBC subset, a pCR was detected in 31% of patients with low TILs, 31% with intermediate, and 50% with high TILs. The authors found a highly statistically significant difference for each subset. In this study, increased TILs were also associated with longer overall survival in TNBC; no associations were detected in HER2-positive patients, while increased TILs were associated with shorter overall survival in luminal–HER2-negative tumors [29]. In the current study, overall survival was not evaluated. Regarding pCR, due to the low sample size, we compared TNBC vs. non-TNBC patients, representing a difference between both studies. We documented a numerically higher NPV for ISCR; however, larger studies are required to evaluate the ISCR, CD3, and CD8 cell densities and the association with endpoints such as pCR, time to progression, and overall survival in early BC patients, as well as PPV and NPV of these biomarkers.

Tertiary lymphoid structures (TLS) are lymph node-like structures that arise in tissues at sites of chronic inflammation [31]. Since the neogenesis of TLS is responsive to chronic inflammation, there is no specific anatomic location or developmental window for TLS induction. TLS have also been detected in human cancers, including thyroid carcinoma, hepatocellular carcinoma, colorectal carcinoma, lung cancer, breast carcinoma, melanoma, prostate cancer, ovarian cancer, and pancreas ductal carcinoma [31]. TLS have been detected in the stroma of up to 60% of BC patients, with the highest frequencies in HER2-positive and TNBC [32,33]. There are, however, significant concerns that TLS cannot be assessed in a reproducible manner by histopathologists using a light microscope and that B- and T-cell immunostains are needed [34].

As reported in ovarian cancer, the higher pre-existing numbers of CD8^+^ TILs in the TME may be activated and mediate their anti-tumor effect due to immunogenic cell death induced by the chemotherapy [35]. Our patients with documented high levels of CD3^+^ and CD8^+^ cells also responded to chemotherapy, suggesting that the presence of these TILs is essential in predicting pCR post-NACT. These findings showed that high IS^®^ is associated with improvement in outcome, as demonstrated previously for colon [36,37] and rectum [38] cancers, reinforcing the importance of a high ISCR, as well as CD3^+^ IM, CD8^+^ IM CD3^+^ CT, and CD8^+^ CT adding relevance to the spatial distribution of these cells. Our data show that the NPV for ISCR was numerically higher compared to CD3^+^ CT, CD8^+^ CT, CD3^+^ IM, and CD8^+^ IM cell counts individually. The PPV was not significant; however, larger prospective studies adequately powered are warranted to establish the utility of the ISCR in this setting.

In conclusion, this is the first study undertaken to evaluate the prognostic value of the CD3^+^ IM, CD8^+^ IM CD3^+^ CT, CD8^+^ CT, and ISCR in BC in the neoadjuvant setting, highlighting not only the prognostic value of this procedure, but also its potential benefit in researching TME cell subsets and their spatial distribution.

## Figures and Tables

**Figure 1 cancers-14-02525-f001:**
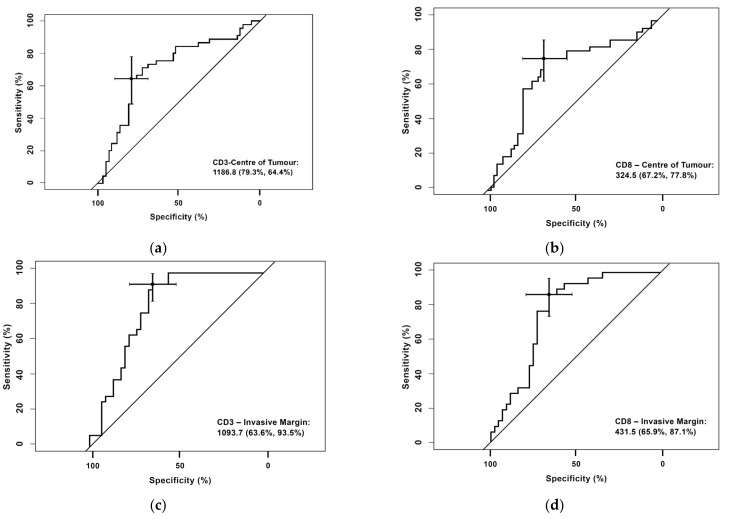
Receiving operating characteristics (ROC) curve for the CD3^+^ and CD8^+^ cell counts in the center of the tumor and the invasive margin in relation to pCR. Data on CD3 IM and CD8 IM were not available on 28 patients. (**a**) ROC Curve: CD3^+^—Center of Tumor; (**b**) ROC Curve: CD8^+^—Center of Tumor; (**c**) ROC Curve: CD3^+^—Invasive Margin; (**d**) ROC Curve: CD8^+^—Invasive Margin.

**Figure 2 cancers-14-02525-f002:**
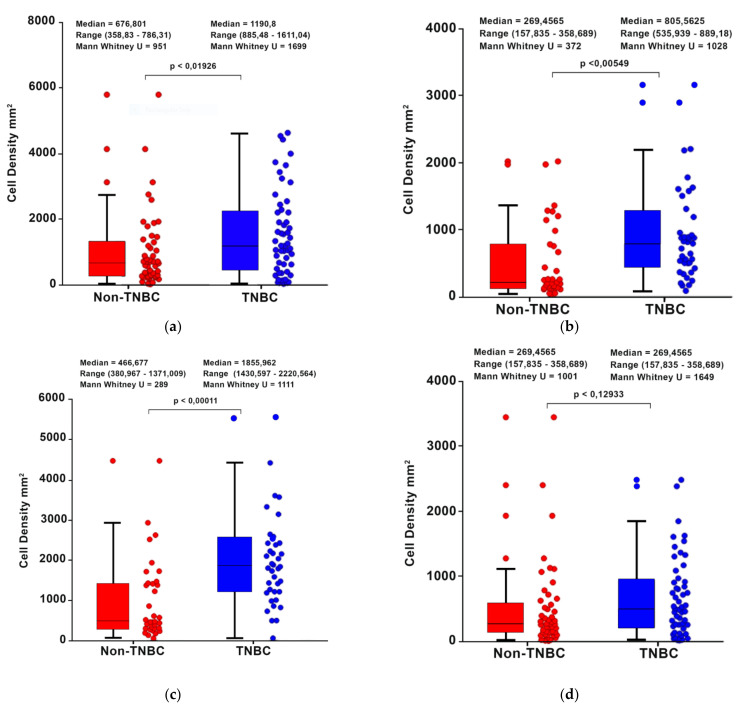
Comparison of T-Cell densities between TNBC vs. Non-TNBC patients. Data on CD3 IM and CD8 IM were not available on 28 patients. (**a**) CD3^+^ cells—Center of Tumor; (**b**) CD3^+^ cells—Invasive Margin; (**c**) CD8^+^ cells—Center of Tumor; (**d**) CD8^+^ cells—Invasive Margin.

**Figure 3 cancers-14-02525-f003:**
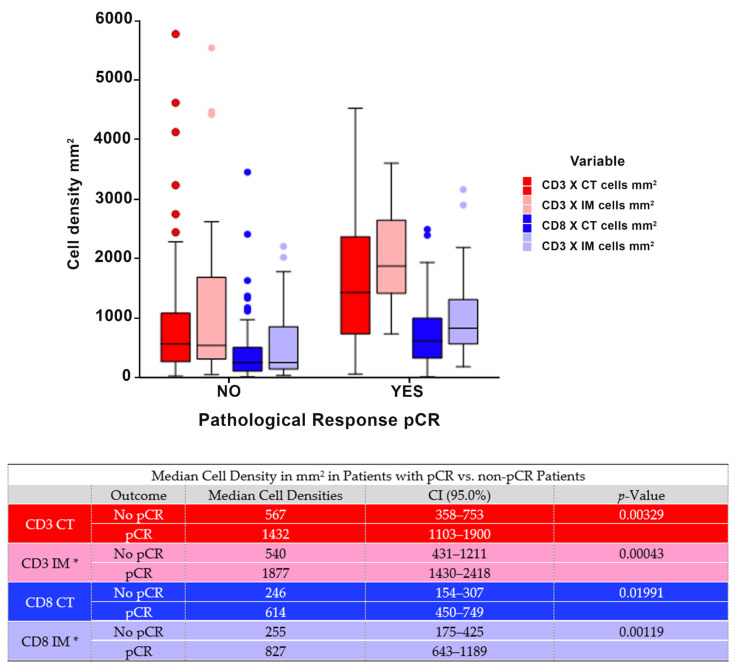
Median cell density in patients with pCR vs. non-pCR patients. * Data on CD3 IM and CD8 IM were not available on 28 patients.

**Figure 4 cancers-14-02525-f004:**
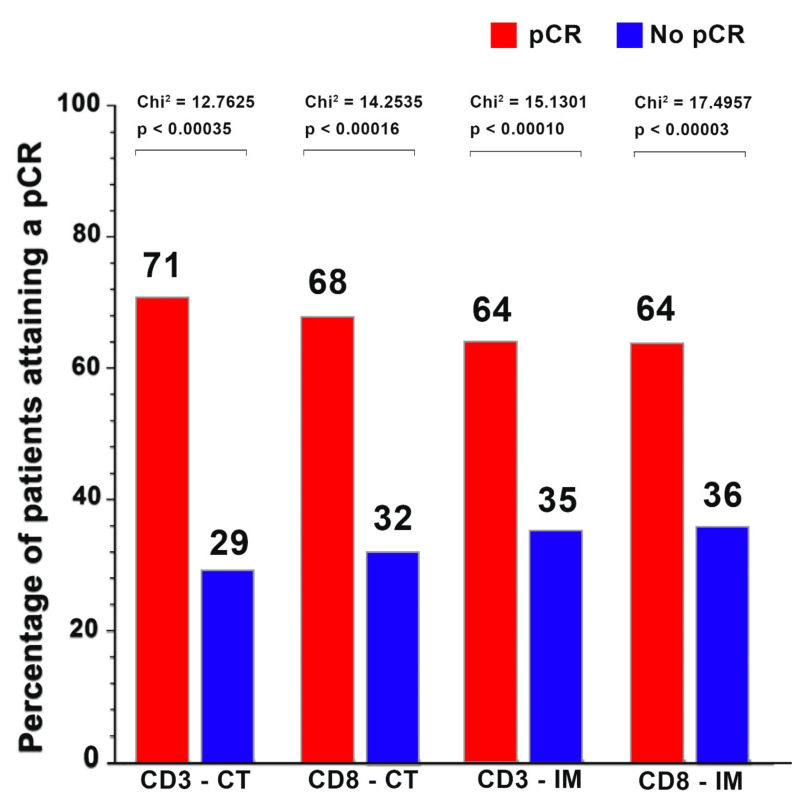
Percentage of patients attaining a pCR by CD3^+^ and CD8^+^ cell densities. Proportion of patients attaining a pCR by CD3 in the center of the tumor (71% vs. 29%), CD8 in the center of the tumor (68% vs. 32%), CD3 in the invasive margin (64% vs. 35%); and CD8 in the Invasive margin (64% vs. 36%). Data on CD3 IM and CD8 IM were not available on 28 patients.

**Table 1 cancers-14-02525-t001:** Patient characteristics.

Total Number of Patients *n* = 103		
Age Median Age (Range/Min–Max) 52 Years (26–84)		
	**Total**	**%**
**Histology**		
Ductal	99	96%
Lobular	2	2%
Other	2	2%
**Menopausal Status**		
Pre-menopausal	41	40%
Post-menopausal	62	60%
**Tumor Size**		
T1	23	22%
T2	65	63%
T3 + T4	15	15%
**Nodal Status**		
Negative	45	44%
Positive	54	52%
Unknown	4	4%
**Stage**		
IA	8	7%
IB	49	48%
IC	10	10%
IIA	1	1%
IIB	26	25%
IIC	7	7%
IIIC	2	2%
**Ki-67 Pre-treatment**		
Ki-67 Median (Range)	40% (5–90%)	
≥40%	51	50%
15–39%	37	35%
<15%	13	13%
Unknown	2	2%
**Molecular subtype**		
Luminal A	9	9%
Luminal B	23	22%
HER-2 Positive	18	18%

**Table 2 cancers-14-02525-t002:** CD3^+^ and CD8^+^ cell counts in the center (103 patients) of the tumor and the invasive margin.

CD3^+^ and CD8^+^ Count	Median Cells/mm^2^	Range (Min Max) Cells/mm^2^
CD3^+^ center of tumor	884	25–5771
CD3^+^ invasive margin *	1409	53–6197
CD8^+^ center of tumor	358	10–3448
CD8^+^ invasive margin *	535	38–3117

* Data on CD3 IM and CD8 IM were not available on 28 patients.

**Table 3 cancers-14-02525-t003:** Percentages of CD3^+^ and CD8^+^ cell counts in the center of the tumor and the invasive margin and Immunoscore CR.

**Cell Density CD3^+^ Center of Tumor**	**Total**	**Percentages**
CD3^+^ CT ≥ 1186.81 cells/mm^2^	40	39%
CD3^+^ CT < 1186.80 cells/mm^2^	63	61%
**Cell density CD3^+^ Invasive Margin ***		
CD3^+^ IM ≥ 1093.71 cells/mm^2^	46	61%
CD3^+^ IM < 1093.70 cells/mm^2^	29	39%
**Cell density CD8^+^ Center of Tumor**		
CD8^+^ ≥ 324.51 cells/mm^2^	55	53%
CD8^+^ < 324.50 cells/mm^2^	48	47%
**Cell density CD8^+^ Invasive Margin ***		
CD8^+^ IM ≥ 431.51 cells/mm^2^	41	55%
CD8^+^ IM < 431.50 cells/mm^2^	34	45%
**Immunoscore CR ***		
0	27	36%
1	4	5%
2	8	11%
3	13	17%
4	23	31%

* Data on the invasive margin were not available on 28 patients.

**Table 4 cancers-14-02525-t004:** Univariate analysis—significant factors associated with pCR.

Variable	pCR	Non-pCR	Fisher’s Exact
**Tumor Size**			
T1	10 (43%)	13 (57%)	0.020
T2	34 (52%)	31 (48%)	
T3 & T4	1 (0.1%)	14 (99.9%)	
**Nodal Status**			
Positive	19 (35%)	35 (65%)	0.024
Negative	24 (53%)	21 (47%)	
**Stage**			
Stage I	6 (67%)	3 (33%)	0.020
Stage IIA	25 (51%)	24 (49%)	
Stage IIB	11 (42%)	15 (58%)	
Stage III	3 (16%)	16 (84%)	
**ER**			
Positive	8 (18%)	37 (82%)	≤0.0001
Negative	37 (64%)	21 (36%)	
**PR**			
Positive	5 (13%)	33 (86%	
Negative	40 (61%)	25 (38%)	≤0.0001
**HER-2**			
Positive	9 (50%)	9 (50%)	0.6070
Negative	36 (42%)	49 (58%)	
**Biological type**			
Luminal	3 (9%)	29 (91%)	≤0.0001
HER-2 Positive	9 (50%)	9 (50%)	
TNBC	33 (62%)	20 (38%)	
**Ki-67**			
≥40%	29 (57%)	22 (43%)	0.0001
15–39%	15 (41%)	22 (59%)	
<15%	0 (0%)	13 (100%)	
**CD3^+^ Cell density at Center of Tumor**			
CD3^+^ CT ≥ 1186.81 mm^2^	27 (68%)	13 (33%)	
CD3^+^ CT < 1186.80 mm^2^	18 (29%)	45 (71%)	0.0002
**CD3^+^ Cell density—Invasive Margin ***			
CD3^+^ IM ≥ 1093.71 mm^2^	29 (63%)	17 (37%)	
CD3+ IM < 1093.70 mm^2^	2 (7%)	27 (93%)	≤0.0001
**CD8^+^ Cell Density at Center of Tumor**			
CD8^+^ CT ≥ 324,51 mm^2^	35 (64%)	20 (36%)	
CD8^+^ CT < 324,50 mm^2^	10 (21%)	38 (79%)	≤0.0001
**CD8^+^ Cell density—Invasive Margin ***			
CD8^+^ IM ≥ 431.51 mm^2^	26 (63%)	15 (37%)	
CD8^+^ IM < 431.50 mm^2^	5 (15%)	29 (85%)	≤0.0001
**Immunoscore CR**			
High	25 (69%)	11 (31%)	≤0.0001
Intermediate	4 (50%)	4 (50%)	
Low	2 (6%)	29 (94%)	

* Data on invasive margin were not available on 28 patients.

**Table 5 cancers-14-02525-t005:** Univariate analysis –significant factors associated with pCR using logistic regression.

Variable	Wald *p*-Value	Odds Ratio	95% CI
**Tumor Size**			
T1 0.0360		1.98	0.04 to 1.32
T2 0.1360		0.51	−1.56 to 0.21
T3 & T4	0.0160	5.49	0.31 to 3.09
**Nodal Status**	0.2460	1.59	1.06 to 2.40
**Stage**			
Stage IIA	0.2905	0.70	−1.01 to 0.30
Stage IIB	0.9951	0.99	−0.76 to 0.76
Stage III	0.0087	3.92	0.35 to 2.38
**ER Status**	<0.0001	1.98	0.50 to 0.86
**PR Status**	<0.0001	1.89	0.46 to 0.81
**HER-2-Status**	<0.0001	0.60	−0.51 to−0.51
**Biological type**			
Luminal—HER-2+	0.0001	3.33	0.62 to 1.78
TNBC—HER-2+	0.0052	0.47	−1.28 to −0.23
**Ki-67** (continuous)	0.0014	0.97	−0.05 to −0.01
**ISCR *** (0 vs. 1 vs. 2 vs. 3 vs. 4)	0.0001	0.42	−1.26 to −0.49
**ISCR *** (0 & 1 & 2 vs. 3 & 4)	<0.0001	12.50	1.40 to 3.65

* Data on Immunoscore Clinical Research were not available on 28 patients.

**Table 6 cancers-14-02525-t006:** Positive predictive values and negative predictive values for CD3^+^ CT, CD8^+^ CT, CD3^+^ IM, CD8^+^ IM, and ISCR for pCR.

	IS (0–1 vs. 2–3–4) (*n* = 75)	CD3 CT (*n* = 75)	CD3 IM (*n* = 75)	CD8 CT (*n* = 75)	CD8 IM (*n* = 75)
**pCR**	41.3	41.3	41.3	41.3	41.3
**PPV**	60.4%	62.2%	63.2%	66.7%	63.4%
95% CI	(45.3–74.2)	(44.8–77.5)	(46.0–78.2)	(49.0–81.4)	(46.9–77.9)
**NPV**	92.6%	79.0%	81.1%	80.0%	85.3%
95% CI	(75.7–99.1)	(62.7–90.4)	(64.8–92.0)	(64.3–91.0)	(68.9–95.1)

## Data Availability

The data of this study is available from the corresponding author upon request.

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
