# Peer review of "Tumor-Infiltrating Lymphocytes (TILs) in Early Breast Cancer Patients: High CD3^+^, CD8^+^, and Immunoscore Are Associated with a Pathological Complete Response"

_cancers, 2022, doi:10.3390/cancers14102525_

Round 1
Reviewer 1 Report
The authors present a well-crafted study assessing the correlation between Immunoscore and pathological complete response in breast cancer patients (triple negative breast cancer vs non-triple negative breast cancer)
A few suggestions for the authors:
- The authors state that CD3+ IM and CD8+ IM score was not available in 28 patients. It would be interesting to state if there are any common factors in these 28 patients (e.g., age, tumor stage, treatment etc) or if this cohort is varied.
- Were any differences in the score observed between patients receiving anthracycline vs taxane vs combination treatment?
- The authors could expand a bit more about this scoring system in their introduction.
- The authors state that CD3+ and CD8+ cells are independent predictors if pCR (Line 259). Similarly, in Line 262 they state that “distribution is important”. It will be important and helpful to graphically represent that:
- A graph comparing pCR vs. nopCR based on CD3 CT vs CD8 CT; CD3 IM vs CD8 IM (This should be done in patients having higher expression of either CD3 or CD8 and then compared to the data from the total cohort to show either could be an independent marker)
- A graph comparing pCR vs. nopCR based on CD3 CT vs CD3 IM; CD8 CT vs CD8 IM. This graph would demonstrate that localization is essential.
Author Response
Reviewer 1
R1. Opening comment
“The authors present a well-crafted study assessing the correlation between Immunoscore and pathological complete response in breast cancer patients (triple negative breast cancer vs. non-triple negative breast cancer).”
“A few suggestions for the authors”
R1. Point 1
“The authors state that CD3+ IM and CD8+ IM scores were unavailable in 28 patients. It would be interesting to state if there are any common factors in these 28 patients (e.g., age, tumor stage, treatment etc) or if this cohort is varied.”
Response
Our response to this comment is shown on page 12, para 2, of the revised manuscript, which states that “with the exception of tumor size, no significant differences were found between the two groups with respect to tumor subtypes, menopausal status, nodal status, and stage of disease.”
R1. Point 2
“Were any differences in the score observed between patients receiving anthracycline vs taxane vs combination treatment?”
Response
As stated on page 13, para 3, of the revision, no meaningful differences were observed in the scores of patients receiving anthracycline vs taxane vs combination treatment due to the small numbers of patients in the anthracycline only and taxane only groups.
R1. Point 3
“The authors could expand a bit more about this scoring system in their introduction.”
Response
This has been done (page 8, para 1 in the introduction section; and page 8, para 3 in the materials and methods section of the revision).
R1. Point 4
“The authors state that CD3+ and CD8+ cells are independent predictors if pCR (Line 259). Similarly, in Line 262 they state that ‘distribution is important’. It will be important and helpful to graphically represent that:
A graph comparing pCR vs. nopCR based on CD3 CT vs CD8 CT; CD3 IM vs CD8 IM (This should be done in patients having higher expression of either CD3 or CD8 and then compared to the data from the total cohort to show either could be an independent marker)
A graph comparing pCR vs. nopCR based on CD3 CT vs CD3 IM; CD8 CT vs CD8 IM. This graph would demonstrate that localization is essential.”
Response
In acknowledging the first of these two points, Figure 4 shows the comparison between pCR vs. non-pCR based on CD3 CT, CD8 CT, CD3 IM, and CD8 IM. We believe that adding an additional figure will confuse the readership. Our opinion is that the best way to show independent factors in this setting is by performing a logistic regression analysis. These comparisons were done and are shown in Table 6.
Again, looking at the second point, we still believe that Figure 4 is the best way to address pCR vs. non-pCR in relationship to CD3 CT, CD8 CT, CD3 IM, and CD8 IM.
Reviewer 2 Report
The authors present a paper on the association between tumor infiltrating lymphocytes (TILs) and complete pathological response in patients receiving neoadjvuant chemotherapy for breast cancer. The paper is well written and comprise mostly sound reasoning. However, it is not very novel and some details needs extra attention before it can be considered for publication.
Abstract: In conclusion the authors state that "These results revealed a significant prognostic role for IS CR in BC". However, they do not investigate the prognostic impact of IS CR in BC. They investigate the association between IS CR and pCR after neoadjvant chemotherapy. This should be clearly stated and amended throughout the paper.7
Introduction: Short. Seems mostly focused on colorectal cancer which is strange since the authors choose to investigate TILs in breast cancer. I suggest to completely rewrite the introduction to reflect the subject of research.
Materials and methods
Patient population: Should include information on how many patients were eligible for the study, how many were excluded and why and how many were included.
IHC: Should at least include a brief description of the IHC procedures. To save space this may be done in a supplementary.
Pathological and clinical assessment: I assume that pCR means pCR in the pathological specimen obtained after definitive surgery. However, that should be clearly stated in this paragraph. Blending clinical and pathological assessment introduced confusion. Since the clinical measures are not used for analyses they may be left out.
Immunoscore CR: It is not stated how the immunoscore was obtained for these patients. Only biopsies were available. How did the authors define the "central tumor"? It seems highly unlikely that this can have been conducted in a reproducible manner - especially since the population is retrospective. If this is not the case, the authors should explain in detail how they adapted the IS CR for use on biopsies.
Statistics: The multivariable model contains far more variables than the number of events allows (number of patients achieving pCR). I addition the IS CD3/CD8 IM and thus the IS CR could not be calculated for 28 patients. It is not stated in the text who these patients were and how many events they comprised. This further weakens the results of the multivariable analyses.
Results
Patients: It seems that the institution had a rather forward selection of patients for neoadjuvant chemotherapy. The study population includes a large proportion of stage I and small tumors. This is not commented, neither in the methods, results nor discussion.
IS CR: From table 4 it seems that the total number of immune cells using CD3 cells/mm2 in either the IM or CT is just as good or better compared to the IS CR in this setting. In fact it seems that the IS CR is just a method to adjust the sensitivity and specificity of the test. In table 6 the authors test some of the significant findings from the univariate analyses in a multivariable model. It appears that they did not investigate their findings for the CD markers alone in this context which they clearly should do.
Discussion: Similar to the introduction, the discussion is mostly concerned with CRC and immunoscore. Dankert et al (ref 28), the largest study to date on TILs in breast cancer patients treated with neoadjvaunt chemotherapy, is only mentioned in a by-sentence. The results of this small study should be compared and discussed according to the Dankert study that is to be considered state of the art in this field.
Author Response
Reviewer 2
R2. Opening comment
“The authors present a paper on the association between tumor infiltrating lymphocytes (TILs) and complete pathological response in patients receiving neoadjuvant chemotherapy for breast cancer. The paper is well written and comprise mostly sound reasoning. However, it is not very novel and some details needs extra attention before it can be considered for publication.”
Response
Although we concede that the issue of high numbers of TILs being a determinant of a favorable clinical outcome in patients with breast cancer, particularly those with the TNBC and HER2+ subtypes, is well recognized, we wish to emphasize that the novelty of our study relates to the apparent utility of the ISCR as an alternative, reproducible, and reliable prognostic adjunct. The study also highlights the principle of researching the TME by looking at various cell subsets and spatial distributions instead of just enumerating the TILs.
R2. Point 1
“Abstract: In conclusion the authors state that ‘These results revealed a significant prognostic role for IS CR in BC.’ However, they do not investigate the prognostic impact of IS CR in BC. They investigate the association between IS CR and pCR after neoadjuvant chemotherapy. This should be clearly stated and amended throughout the paper. 7.”
Response
This point is well taken and has been implemented throughout the revision.
R2. Point 2
“Introduction: Short. Seems mostly focused on colorectal cancer which is strange since the authors choose to investigate TILs in breast cancer. I suggest to completely rewrite the introduction to reflect the subject of research.”
Response
The “Introduction” has been restructured as suggested with an increased focus on the ISCR as a prognostic tool in relationship to pCR.
R2. Point 3
“Materials and methods
Patient population: Should include information on how many patients were eligible for the study, how many were excluded and why and how many were included.”
Response
This information has been included on page 8, para 2 of the revision. We included consecutive patients with a complete data set and available blocks.
R2. Point 4
“IHC: Should at least include a brief description of the IHC procedures. To save space this may be done in a supplementary.”
Response
A brief description of the immunohistochemistry has been included in the body of the text (page 8, para 4, and page 9, para1).
R2. Point 5
“Pathological and clinical assessment: I assume that pCR means pCR in the pathological specimen obtained after definitive surgery. However, that should be clearly stated in this paragraph. Blending clinical and pathological assessment introduced confusion. Since the clinical measures are not used for analyses they may be left out.”
Response
Your interpretation is correct and somewhat similar to your point 1. This has been clarified on page 9, para 2, of the revision. Additionally, the statement related to the clinical assessment is a valid point and has been deleted.
R2. Point 6
“Immunoscore CR: It is not stated how the immunoscore was obtained for these patients. Only biopsies were available. How did the authors define the ‘central tumor’? It seems highly unlikely that this can have been conducted in a reproducible manner - especially since the population is retrospective. If this is not the case, the authors should explain in detail how they adapted the IS CR for use on biopsies.”
Response
This issue has been clarified on pages 9, para 2, and 3; of the revision.
R2. Point 7
“Statistics: The multivariable model contains far more variables than the number of events allowed (number of patients achieving pCR). In addition the IS CD3/CD8 IM and thus the IS CR could not be calculated for 28 patients. It is not stated in the text who these patients were and how many events they comprised. This further weakens the results of the multivariable analyses.”
Response
This information has been included on page 11, para 1, going into page 12 of the revision.
“An analysis of the characteristics of the 75 patients with a documented IM score relative to the 28 patients without a documented IM score was undertaken to detect potential differences between the two groups. Among the 75 patients with known IM scores, there were 40 (53%) TNBC patients, 23 (31%) luminal patients, and 12 (16%) HER2-positive patients. When compared to the distribution of the molecular subtypes of the full cohort of 103 patients with the 75 patients with known IM ISCR, we did not detect any significant difference between the two groups (TNBC 51% vs. 53%, luminal subtype 31% vs. 31% and HER2-positive 18% vs. 16%; chi2 = 0.547, DF = 2, p < 0.7607).
No significant difference in terms of menopausal status (pre-menopausal vs post-menopausal; chi2 0.004, p < 0.9475), nodal status (negative vs positive; chi2 1.001, p < 0.3170), stage (stage I vs IIA vs IIB vs III; chi2 2.265, p < 0.5192) or molecular sub-type (TNBC vs luminal vs HER2 status; chi2 0.547, p < 0.7607) was found between the two cohorts, with the exception of tumor size (T1 vs T2 vs T3 combined with T4; chi2 9.358, p < 0.0093).”
R2. Point 8
“Results
Patients: It seems that the institution had a rather forward selection of patients for neoadjuvant chemotherapy. The study population includes a large proportion of stage I and small tumors. This is not commented, neither in the methods, results nor discussion.”
Response
This issue has been addressed on page 8, paras 2 of the revision.
“It is an institutional policy to use neoadjuvant chemotherapy for most patients with HER2-positive and TNBC.”
R2. Point 9
“IS CR: From table 4 it seems that the total number of immune cells using CD3 cells/mm2 in either the IM or CT is just as good or better compared to the IS CR in this setting. In fact it seems that the IS CR is just a method to adjust the sensitivity and specificity of the test. In table 6 the authors test some of the significant findings from the univariate analyses in a multivariable model. It appears that they did not investigate their findings for the CD markers alone in this context which they clearly should do.”
Response
As requested, the finding in relation to the inclusion of the CD markers alone was included in the multivariate analysis and reported in Table 6.
R2. Point 10
“Discussion: Similar to the introduction, the discussion is mostly concerned with CRC and immunoscore. Dankert et al (ref 28), the largest study to date on TILs in breast cancer patients treated with neoadjuvant chemotherapy, is only mentioned in a by-sentence. The results of this small study should be compared and discussed according to the Dankert study that is to be considered state of the art in this field.”
Response
A discussion of the Denkert study (reference 29 in the revision) in relation to the study under review has been included on page 16, para 1 of the revision.
Reviewer 3 Report
The manuscript by Rapoport et al the use of immunoscore as a prognostic tool in both invasive margin and center of the tumor in triple-negative breast cancer (TNBC). Most of the results have already been shown in other publications but this study focuses on a more quantitative methodology by using immunoscore which could provide a more objective quantification of TILs
- There is at least one article using immunoscore in TNBC in open access that has not been referenced.
https://www.journalaorj.com/index.php/AORJ/article/view/30127
Also the conclusions of this publication should be discussed with respect the presented manuscript and the real advancement of the presented manuscript highlighted.
- Authors claim that “To our knowledge, our study is the first to differentiate CT and IM when analyzing the influence of TILs on prognosis after NACT in early breast cancer”. This might be true; however they should discuss extensively the usefulness of such differentiation and the practical consequences given it might be the novelty of the article.
- I guess the number of samples limits the segregation of data by subtypes. However it is known that luminal BC do not respond to neoadjuvant treatment and hardly produce pCR. In fact table 4 in the biological type section suggests this. Authors should elaborate this issue.
- Authors should clarify to which subtypes correspond the data on CD3 IM and CD8 IM which are not available on 28 patients....were those of predominant luminal subtype ?. Perhaps this should be discussed. In fact, I believe Figures 2 and 4 should be separated by subtypes.
- Also for this group of patients specifically, how do CD3 and CD8 appear in the center of tumor ?.
- The text is a bit repetitive with respect the information provided in the tables. Authors shoud try to elaborate the meaning of the obtained results in the text.
Minor points:
- Table one should appear as soon as possible in the manuscript
- Line 50. TNM abbreviation should be defined.
- Fig2 2 page 7. Fig2A and 2B miss the x axis and legend. Fig 2B, 2C and 2D show outliers in the opposite colour for TNBC.
- In fact Figure 2 should be elaborated for the different subtypes. I very much doubt the luminals follow this tendency.
- Please correcto typos in table 4.CD3 Cell density IM missing + in CD3. Also CD8+ at TC has contradiction in the intervals, both are greater than or equal to.
- Line 256. Given the wide range of ages in the patients, why do authors claim they analyze the influence of TILs on prognosis after NACT in early breast cancer ?.
- How do the wide range of patient age influence the interpretation of the results ?
- Line 262. “This illustrates that lymphocyte distribution within the tumor mass is more important than just the presence of TILs “ Please clarify what is important for.
- Line 267. Can the authors clarify to what extent references 18-24 lead to inconsistent results ?.
Author Response
Reviewer 3
R3. Opening comment
“The manuscript by Rapoport et al. the use of immunoscore as a prognostic tool in both invasive margin and center of the tumor in triple-negative breast cancer (TNBC). Most of the results have already been shown in other publications but this study focuses on a more quantitative methodology by using immunoscore which could provide a more objective quantification of TILs.”
R3. Point 1
“There is at least one article using immunoscore in TNBC in open access that has not been referenced.
https://www.journalaorj.com/index.php/AORJ/article/view/30127
Also the conclusions of this publication should be discussed with respect the presented manuscript and the real advancement of the presented manuscript highlighted.”
Response
We wish to point out that the article to which, this reviewer refers (Tshuma et al., Asian Oncology Research Journal, 2021) is, in fact, a review in which the authors advocate, as opposed to investigating, the potential utility of the immunoscore as a prognostic tool in breast cancer.
R3. Point 2
“Authors claim that ‘To our knowledge, our study is the first to differentiate CT and IM when analyzing the influence of TILs on prognosis after NACT in early breast cancer’. This might be true; however they should discuss extensively the usefulness of such differentiation and the practical consequences given it might be the novelty of the article.”
Response
This issue has been addressed throughout the discussion of the revision.
R3. Point 3
“I guess the number of samples limits the segregation of data by subtypes. However it is known that luminal BC do not respond to neoadjuvant treatment and hardly produce pCR. In fact table 4 in the biological type section suggests this. Authors should elaborate this issue.”
Response
This issue has been addressed on page 10, para 16 of the revision.
“The pCR rate for the entire cohort was 44% (luminal 9%, TNBC 62%, and HER2-positive 50%).”
R3. Point 4
“Authors should clarify to which subtypes correspond the data on CD3 IM and CD8 IM which are not available on 28 patients ... were those of predominant luminal subtype?. Perhaps this should be discussed. In fact, I believe Figures 2 and 4 should be separated by subtypes.”
Response
Data from the 28 patients were included by subtype on page11 para 3 and page 12, para 1.
An analysis of the characteristics of the 75 patients with a documented IM score relative to the 28 patients without a documented IM score was undertaken to detect potential differences between the two groups. Among the 75 patients with known IM scores, there were 40 (53%) TNBC patients, 23 (31%) luminal patients, and 12 (16%) HER2-positive patients. When compared to the distribution of the molecular subtypes of the full cohort of 103 patients with the 75 patients with known IM ISCR, we did not detect any significant difference between the two groups (TNBC 51% vs. 53%, luminal subtype 31% vs. 31% and HER2-positive 18% vs. 16%; chi2 = 0.547, DF = 2, p < 0.7607).
Regards Figures 2 and 4, we subdivided into TNBC vs. non-TNBC. We plan a follow-up manuscript with further sub-analysis, as this manuscript already contains a substantial amount of data.
R3. Point 5
“Also for this group of patients specifically, how do CD3 and CD8 appear in the center of tumor?”
Response
This terminology refers to TILs, which are alongside (adjacent) to the tumor. This point was described in the Patients and Methods sections of the revision.
R3. Point 6
“The text is a bit repetitive with respect the information provided in the tables. Authors should try to elaborate the meaning of obtained results in the text.”
Response
In addressing this point, we have attempted to interpret, as opposed to simply repeating, the results shown in the tables in the text.
R3. Minor Points
- Table one should appear as soon as possible in the manuscript.
- Line 50. TNM abbreviation should be defined.
- Fig2 2 page 7. Fig 2A and 2B miss the x axis and legend. Fig 2B, 2C and 2D show outliers in the opposite colour for TNBC.
- In fact Figure 2 should be elaborated for the different subtypes. I very much doubt the luminals follow this tendency.
- Please correct typos in table 4. CD3 Cell density Im missing + in CD3. Also CD8+ at TC has contradiction in the intervals, both are greater than or equal to.
- Line 256. Given the wide range of ages in the patients, why do authors claim they analyze the influence of TILs on prognosis after NACT in early breast cancer?.
- How do the wide range of patient age influence the interpretation of the results?
- Line 262. “This illustrates that lymphocyte distribution within the tumor mass is more important than just the presence of TILs” Please clarify what is important for.
- Line 267. Can the authors clarify to what extent references 18-24 lead to inconsistent results?
Response
- Table 1 is now mentioned at an earlier point in the text in the revision (page 8, para 2).
- The abbreviation TNM has been explained (page 7, para 2).
- In Figures 2A – 2D, it is only the horizontal line (x-axis) on Figure 2A that is missing. This has been rectified in the revision.
- Subtypes? We already responded to this point (under reviewer 3 point 4 as above).
- In Table 4, CD3 has been changed to CD3+, and the lower CD8+ symbols changed from ≥ to <.
- We analyzed the data by menopausal status and not by age. To analyze the data also by age would almost be repetitive.
- Please refer to point: Menopausal status as opposed to age.
- This has been explained on page 15, para 1, of the revision, meaning that proximity to the tumor rather than simply the random presence of TILs in the TME is probably critical.
- The word inconsistent has been deleted, and greater emphasis is placed on differences between references 20-26 and the ISCR.
Round 2
Reviewer 1 Report
The revised manuscript reads well. I recommend acceptance.
Author Response
The revised manuscript reads well. I recommend acceptance
Reviewer 2 Report
The authors present a partly revised version of their manuscript describing the association between tumor infiltrating lymphocytes (TILs) and complete pathological response in patients receiving neoadjuvant chemotherapy for breast cancer. Some suggestions from the previous review round has been addressed. Nevertheless, the manuscript still contains major flaws. New comments are in normal text. Comments not properly addressed from the previous review round are in italics. In addition, the references the authors provide for their changes are not corresponding to the document sent to the reviewers.
Abstract: In conclusion the authors state that ‘These results revealed a significant prognostic role for IS CR in BC.’ However, they do not investigate the prognostic impact of IS CR in BC. They investigate the association between IS CR and pCR after neoadjuvant chemotherapy. This should be clearly stated and amended throughout the paper. In their response the authors claim to have addressed this point. However, this is not the case in most parts of the manuscript (easiest to spot in the abstract where it was originally pointed out...)
Statistics: The multivariable model contains far more variables than the number of events allowed (number of patients achieving pCR). In addition the IS CD3/CD8 IM and thus the IS CR could not be calculated for 28 patients. It is not stated in the text who these patients were and how many events they comprised. This further weakens the results of the multivariable analyses. This point is partly addressed in the revised version of the manuscript. The authors does not address the power of the multivariable model. Since the study is clearly to underpowered to do a "real" multivariable analysis these results should be omitted entirely.
IS CR: From table 4 it seems that the total number of immune cells using CD3 cells/mm2 in either the IM or CT is just as good or better compared to the IS CR in this setting. In fact it seems that the IS CR is just a method to adjust the sensitivity and specificity of the test. In table 6 the authors test some of the significant findings from the univariate analyses in a multivariable model. It appears that they did not investigate their findings for the CD markers alone in this context which they clearly should do. In the revised manuscript the authors include the CD markers in the multivariable analysis. This is not properly addressing the results. The study is not powered to do a multivariable anlysis and this should be omitted from the manuscript entirely. The CD markers should be presented in the univariable table in order to compare their prognostic potential alongside the IS CR.
Discussion: Similar to the introduction, the discussion is mostly concerned with CRC and immunoscore. Dankert et al (ref 28), the largest study to date on TILs in breast cancer patients treated with neoadjuvant chemotherapy, is only mentioned in a by-sentence. The results of this small study should be compared and discussed according to the Dankert study that is to be considered state of the art in this field. The authors does not provide an adequate comparison that highlights the benefits of their study compared to the Denkert study.
Discussion: The authors does not comment on how their data may be used in a clinical setting? What is important? NPV? PPV? Other parameters?
Author Response
Abstract: In conclusion the authors state that ‘These results revealed a significant prognostic role for IS CR in BC.’ However, they do not investigate the prognostic impact of IS CR in BC. They investigate the association between IS CR and pCR after neoadjuvant chemotherapy. This should be clearly stated and amended throughout the paper. In their response the authors claim to have addressed this point. However, this is not the case in most parts of the manuscript (easiest to spot in the abstract where it was originally pointed out...)
Our reply: We have included the wording “pCR following neoadjuvant chemotherapy” throughout the revised manuscript.
Statistics: The multivariable model contains far more variables than the number of events allowed (number of patients achieving pCR). In addition the IS CD3/CD8 IM and thus the IS CR could not be calculated for 28 patients. It is not stated in the text who these patients were and how many events they comprised. This further weakens the results of the multivariable analyses. This point is partly addressed in the revised version of the manuscript. The authors does not address the power of the multivariable model. Since the study is clearly to underpowered to do a "real" multivariable analysis these results should be omitted entirely.
Our reply: The multivariate analysis has been removed in the revised manuscript.
IS CR: From table 4 it seems that the total number of immune cells using CD3 cells/mm2 in either the IM or CT is just as good or better compared to the IS CR in this setting. In fact it seems that the IS CR is just a method to adjust the sensitivity and specificity of the test. In table 6 the authors test some of the significant findings from the univariate analyses in a multivariable model. It appears that they did not investigate their findings for the CD markers alone in this context which they clearly should do. In the revised manuscript the authors include the CD markers in the multivariable analysis. This is not properly addressing the results. The study is not powered to do a multivariable anlysis and this should be omitted from the manuscript entirely. The CD markers should be presented in the univariable table in order to compare their prognostic potential alongside the IS CR.
Our reply: The multivariate analysis has been removed in the revised manuscript. Univariate CD markers are shown in tables 4 and 5 in the revised manuscript.
Discussion: Similar to the introduction, the discussion is mostly concerned with CRC and immunoscore. Dankert et al (ref 28), the largest study to date on TILs in breast cancer patients treated with neoadjuvant chemotherapy, is only mentioned in a by-sentence. The results of this small study should be compared and discussed according to the Dankert study that is to be considered state of the art in this field. The authors does not provide an adequate comparison that highlights the benefits of their study compared to the Denkert study.
Our reply: The Denkert study was expanded and covered in detail and compared to our fings in the revised manuscript.
Discussion: The authors does not comment on how their data may be used in a clinical setting? What is important? NPV? PPV? Other parameters?
Our reply: We addressed this issue in detail in the discussion and under pitfalls of the current study and future directions, including NPV and PPV
Reviewer 3 Report
Most concerns have been addressed
Author Response
Most concerns have been addressed
Round 3
Reviewer 2 Report
The authors have now complied with most of my concerns. Two points remain
1: There is still some uncertainty of when the analyzed tissue was collected. In the abstract the authors write
Pre-treatment tumor biopsies were immune-stained for
CD3+ and CD8+ T-cell markers. Quantitative analysis of these cells in different tumor
locations was performed using computer-assisted image analysis.
While in the material and methods they write:
For each patient, a paraffin-embedded tumor block from the Center of the Tumor
(CT) and the Invasive Margin (IM) was selected.
Was the tissue collected before or after final surgery? If the IM and CT was defined on pre chemo biopsies - how was this done? If the final surgical specimen was used to define IM and CT it should be discussed how this may skew results.
2: The discussion still lacks a section where the authors compare their results for the CD markers with those of the immunoscore. They should provide arguments for the introduction of a complex score that was unable to score 25% of patients compared to simple enumeration of CD3 and/or CD8
Author Response
Date: 09-May-2022
Dr. J. Bogdanović
Assistant Editor: “CANCERS“
MDPI
Belgrade
Dear Dr. Bogdanović
Re: Manuscript ID: cancers-1668775: Tumor-Infiltrating Lymphocytes (TILs) in Early Breast Cancer Patients: High CD3+, CD8+, and Immunoscore are Associated with Pathological Complete Response.– Revised
Herewith the revised version of our manuscript (ID: cancers-1668775), which has been amended in accordance with the recommendations of the expert reviewer number 2 as follows:
1: There is still some uncertainty of when the analyzed tissue was collected. In the abstract the authors write
Pre-treatment tumor biopsies were immune-stained for CD3+ and CD8+ T-cell markers. Quantitative analysis of these cells in different tumor locations was performed using computer-assisted image analysis.
While in the material and methods they write:
For each patient, a paraffin-embedded tumor block from the Center of the Tumor
(CT) and the Invasive Margin (IM) was selected.
Was the tissue collected before or after final surgery? If the IM and CT was defined on pre chemo biopsies - how was this done? If the final surgical specimen was used to define IM and CT it should be discussed how this may skew results.
Response: As clearly stated throughout in the revised manuscript, all tissue was taken prior to the initiation of NACT. I would like to point out to the reviewer that the standard of care is to perform a pre-treatment biopsy. Additionally, it is impossible to analyze tissue from samples at the time of surgery for two reasons. 1) If the patient attains a pCR, there will be no tumor cells available at the time of surgery. 2) On the patients who did not attain a pCR, ample data shows that the tumor microenvironment changes following NACT, making it impossible to interpret this information.
2: The discussion still lacks a section where the authors compare their results for the CD markers with those of the immunoscore. They should provide arguments for the introduction of a complex score that was unable to score 25% of patients compared to simple enumeration of CD3 and/or CD8
Response We added additional statistical analyses and showed numerically higher NPV. We stated that further studies should be done to confirm these findings. These studies should be well designed, prospective, and adequately powered. To accommodate this change, we have also made a minor change to the title of the revised manuscript.
In conclusion, we trust that our responses to the expert reviewer, are satisfactory and that the revised version of our manuscript is now acceptable for publication in “CANCERS.”
With thanks & best regards,
Yours sincerely,
Prof B.L. Rapoport.
